# Invited perspectives: Challenges and step changes for natural hazards - perspectives from the German Committee for Disaster Reduction (DKKV)

Benni Thiebes[1], Ronja Winkhardt-Enz[1], Reimund Schwarze[1,2], Stefan Pickl[1,3]

[1]German Committee for Disaster Reduction (DKKV), Bonn, 53115, Germany
[2]Helmholtz Zentrum für Umweltforschung – UFZ, Leipzig, 04318, Germany
[3]Universität der Bundeswehr München, Neubiberg, 85577, Germany

*Correspondence to*: Benni Thiebes (benni.thiebes@dkkv.org)

**1 The German Committee for Disaster Reduction | Deutsches Komitee für Katastrophenvorsorge e.V.**

DKKV is the largest platform for disaster risk reduction in Germany and has been promoting prevention and disaster risk reduction in science, practice and in politics since the beginning of the International Decade for Natural Disaster Reduction in 1990. The overarching leitmotiv of DKKV is the strengthening of societal resilience and fostering the exchange and knowledge transfer between research, operative actions, the policy level, and the public. DKKV acts as an umbrella organization for German institutions and experts in the field of disaster risk reduction and functions as an intermediary to national and international networks and initiatives. Students and recent graduates are involved in the network and can actively participate in DKKV activities for a reduced membership fee. We hereby aim to empower the Young Professionals and invite them to contribute to disaster risk reduction through the use of science, engineering, technology, and innovation through participation.

As described in figure 1, DKKV is active in three fields, i.e., (1) networking and fostering exchange between experts from science, practice, administration, and politics; (2) consulting of decision-makers to provide best-practices and actionable advice to stakeholders; and (3) knowledge management and transfer addressing civil society and the general public to promote prevention measures, e.g. via our website, flyers and public lectures and workshops. DKKV is an active member of national and international networks, e.g., it acts as the national focal point for IRDR (Integrated Research on Disaster Risk) and GNDR (Global Network of Civil Society Organisations for Disaster Reduction), is observer organisation for UNFCCC (United Nations Framework Convention on Climate Change), is active member of E-STAG (European Science and Technology Advisory Group), and contributes to networks such as Risk KAN (Knowledge Action Network on Emergent Risks and Extreme Events) and DERN (Disaster and Emergency Research Network). DKKV frequently contributes to research projects such as MATRIX (FP7), ESPREssO (H2020), and PLACARD (H2020). Within research projects, DKKV generally aims to contribute to communication and dissemination as well as to tasks which require to synthesise research results for different stakeholders and communities. At the same time, these project participations allow DKKV to extend and strengthen

its network, develop new methods and approaches (such as serious games), and provide funding for activities that create synergies between project tasks and activities that lie in the field of our institutional goals.

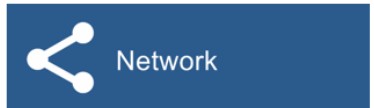 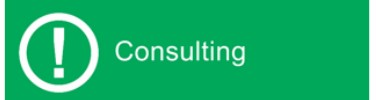 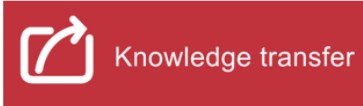

**Foster exchange and synergies**
*for experts*

- Coordinate project consortia
- Workshops and statements
- Science to practice
- Website, newsletter, social media
- Co-organization of national congress on disaster prevention
- Young professional award

**Provide actionable advice**
*for decision-makers*

- Best-practices and lessons-learned
- Accompany international frameworks
- Develop risk and crisis scenarios
- Synthesis reports
- Research and consulting projects

**Raise awareness**
*for general public*

- Knowledge management
- Documentation of state of the art
- Information and actionable advice for prevalent risks
- Moderation of dialogues on future topics
- Public lectures
- National-scale resilience reports

*Figure 1: Main working fields of DKKV are networking, consulting, and knowledge transfer*

## 2 Challenges for future research and resilience building

The worldwide challenge of the present as well as the future is to navigate the global community to a sustainable and secure future. Humanity is increasingly facing multiple risks under more challenging conditions. The continuation of climate change and the ever more frequent occurrence of extreme, multi-hazard, and cascading events are interacting with increasingly complex and interconnected societies. Natural and technological hazards leading to economic and non-economic losses call for further research to assist policy makers and public administration with evidence-based, precautionary, and actionable information and advice. It is therefore important for research to focus more on overarching and synergetic approaches, such as integrating disaster risk reduction (DRR), climate change adaptation (CCA) and complexity science to foster resilience. The strengthening of resilience, i.e. a system's (i) absorptive, (ii) adaptive, (iii) anticipative, (iv) preventive, and (v) transformative capacities to respond to risks and crises (UNDP, 2020) is at the core of key international frameworks such as the Sustainable Development Goals (SDGs) (United Nations, 2015b), the Paris Agreement on Climate Change (United Nations, 2015a), the Sendai Framework for Disaster Risk Reduction 2015-2030 (SFDRR) (UNDRR, 2015), and the New Urban Agenda (UN Habitat, 2016). Thus, creating synergies in research and practice is key to strengthening resilience. In relation to the flood disaster in July 2021, DKKV has provided a platform for researchers to enable synergies between efforts and to avoid duplication of research. In addition to an expert excursion to the Ahr valley, a transdisciplinary round table discussion was organised as well as an open to everyone online event, where researcher working on the flood could exchange on re-

search ideas and to align and complement research activities. Such bottom-up approaches represent and important mean to facilitate inclusive, integrative, transdisciplinary, and synergetic collaboration.

## 3 Step changes and research needs according to the priorities of the SFDRR

Targeting the Sendai priorities, a vision paper, DKKV contributed to, (Zuccaro et al., 2018) put forward as a main outcome of the ESPREssO project (www.espressoproject.eu) and identified research needs and emerging cross-cutting issues for future research in DRR and CCA. This is also in line with the upcoming IRDR Research Agenda for Global Science in Support of Risk-Informed Sustainable Development and Planetary Health (IRDR International Conference 2021).

*Table 1: Sendai Priorities with respective step changes*

| Understanding disaster risk | Strengthening disaster risk governance to manage disaster risk | Investing in disaster reduction for resilience | Enhancing disaster preparedness for effective response, and to "Build Back Better" in recovery, rehabilitation, and reconstruction |
|---|---|---|---|
| Interdisciplinary research with a systemic, integrative perspective and co-creation of knowledge | Strengthening of system's (i) absorptive, (ii) adaptive, (iii) anticipative, (iv) preventive, and (v) transformative capacities | Overcoming the implementation gap in DRR and CCA | Improved early warning systems, both technologically and organizationally, to enhance disaster preparedness |
| Improved data management and information updates; harmonised data, protocols, and procedures | improved knowledge-based decision-making and transboundary cooperation along the entire Disaster Risk Management cycle | implementation to disseminate the best practices experimented | Greater national and international visibility of success stories of effective response and Build Back Better" in recovery, rehabilitation, and reconstruction |
| efficient communication and dissemination platforms | Implementation of community governance with continuous consultation with impacted communities. | country-specific and international priorities in a balanced way | improved communication to the general public |

Several step changes in terms of improved understanding of disaster risk emerge, as displayed in the table 1 above. As risk is composed of the elements of hazard, vulnerability, and exposure, interdisciplinary research with a systemic and integrative perspective is needed. In this regard, scientifically addressing all elements of risk is fully consistent with the societal need for improved risk and impact assessments as well as improved data for decision-making on a resilient future. Improved data entails both harmonization of data collection, its effective management and continuous updating. These are fundamental for more detailed and advanced simulations and impacts assessments (e.g., Kaewunruen et al., 2021; Losier et al., 2019; Peichl et al., 2021). Based on co-creation of knowledge with all involved actors and communities can increase the overall awareness and preparedness to risk through efficient communication and dissemination platforms. A multi-level approach and data-driven decision-making process is needed to further strengthen risk governance to manage disaster risk. This implies the effective implementation of community governance with continuous consultation with impacted communities at the local, national, and international level. Moreover, at the policy level, existing synergies between DRR, CCA, and SDGs should be

enhanced and expanded, with the aim to improve the legal framework for sustainable and holistic decision-making. This applies to all components of the disaster risk management cycle. Equally important is an international transboundary cooperation for long-term joint preparedness, adaptation, and mitigation of risk. The same applies to joint emergency management, response coordination and a sustained recovery. DKKV for example is fostering European exchange on transformations towards a sustainable and resilient future by organising events with national platforms from France (AFPCN) or Austria (DCNA).

In this regard, enhanced partnerships and improved legal frameworks, procedures, and tools represent an important step change for knowledge-based decision-making. An additional step change must be geared toward investing in DRR for resilience. Due to the multiple emerging risk conditions societies are facing, it is crucial to overcome the implementation gap in DRR and CCA in order to strengthen resilience. Best-practices tested through community action in DRR and CCA need greater national and international visibility through more effective dissemination and application. Investing in scientific research and production of knowledge on natural hazards, vulnerabilities, and exposure will help anticipate and better prepare for increased risk situations and increase resilience. This is fully consistent with enhancing disaster preparedness for effective response and to "Build Back Better" in recovery, rehabilitation, and reconstruction.

To optimize the available resources in an integrative way, a widespread adoption of "Build Back Better" principles is essential, to allow greater flexibility of measures and actions for DRR and CCA. Thus, reconstruction after major disaster events, such as the floods in Germany in 2021, requires scientific advice and assessment of future risks and impacts to allow for risk informed and participative decision-making. As a consequence, two projects on sustainable reconstruction were recently funded by the German Ministry for Research and Education. Furthermore, early warning systems need a step change, both technologically and organizationally, to enhance disaster preparedness. Recent positive developments in Germany include e.g., lowering of in the legal barriers for the national government to provide coordination in regional and state-level disaster events as well as the upcoming implementation of a cell broadcasting alert system. Communication with the public represents a crucial factor, which needs to be improved, considering societal values and human behaviours while protecting vulnerable communities and ensuring the transparency in decision-making. Serious games have proven to be an effective tool for the communication of complex issues in a playful environment while at the same time helping to break up silo thinking. DKKV uses RAMSETE III (Booth et al., 2020; Fleming et al., 2020) and other gaming approaches in workshops for decision-makers and researchers to communicate the challenges of decision making under uncertain risk conditions.

As climate change and disasters do not respect political borders, developing transboundary international coordination and cooperation mechanisms that take into account emerging risks increases the overall resilience.

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
