# Peer review of "Invited perspectives: Challenges and step changes for natural hazards - perspectives from the German Committee for Disaster Reduction (DKKV)"

_Natural Hazards and Earth System Sciences, 2021_

## Author Response (AR1)

**Point-by-point reply to the comments**

Dear editor, dear reviewers,

Many thanks for your constructive comments on our manuscript. Here, we would like to provide the requested point-by-point reply to your comments to highlight how we have addressed your feedback during the revision. Additionally, we have provided the manuscript as a marked-up version as well as a clean version.

A general shortcoming of our initial submission described by both reviewers was the lack of concrete examples. We have thus tried to provide more illustrative examples of our activities.

Best regards,

Benni Thiebes et al.

**Comments by editor**

Dear Dr. Thiebes and co-authors,
thank you very much for your responses to the two reviews. The discussion is now closed, and I would like to give you some advice on the revision you are now invited to submit, and which will go to the reviewers again.
With respect to the extension of the paper and the foreseen length, I agree that a modest extension with figures and a concrete examples is beneficial, so that you can proceed in this respect. This does not mean that I would suggest a major extension of the manuscript.

Reply: Thank you for your understanding that addressing the comments raised by the reviewers goes along with a modest extension of the manuscript. We have tried to keep it brief and hope that the revised version is acceptable.

With respect to the specific comment to line 27 of reviewer 2, my understanding is not that a specific activity (serious gaming) is what is desired, but that your typical activities in the projects are named. Your activities may be specific compared to the general goals of individual research projects. Your role is, in my opinion, more interesting for the readers. This is in line with the reviewer's comment that "It is difficult … to picture the exact tasks and responsibilities of the DKKV."

Reply: Thank you for this clarification. We have added the following sentence:

*"Within research projects, DKKV generally aims to contribute to communication and dissemination as well as to tasks which require to synthesise research results for different stakeholders and communities. At the same time, these project participations allow DKKV to extend and strengthen its network, develop new methods and approaches*

*(such as serious games), and provide funding for activities that create synergies between project tasks and activities that lie in the field of our institutional goals."*

I am looking forward to your revised manuscript.

Best regards,
Uwe Ulbrich

Reply: Again, thank you for your support.

**Comments by reviewer 1**

This article is an invited perspective for the special issue "Perspectives on challenges and steps changes for addressing natural hazards". The aim is to present some insights on the challenges of natural hazards based on the experience collected within the German Committee for Disaster Reduction (DKKV). The article is structured in three squeezed sections: 1) presentation of DKKV, 2) future research and resilience buildings, 3) step changes and needs according to the Sendai Framework for Disaster Risk Reduction.

Overall the work is acceptable, well written, and in line with the special issue goals; however, after reading it the feeling is that something is missing, especially in the presentation of the DKKV experience that could be enlarged:

a) one or two figures (diagrams) illustrating the content of each section (especially 1 and 3) could help the reader to better understand the framework of DKKV first but also the road map of step changes and needs for natural hazards management.

Reply: We have added a figure on the fields of activities of DKKV with some examples of what these encompass. An additional table aims to give a better overview on where step changes are suggested in relation to the Sendai priorities.

b) the description of some practical example/storytelling (maybe 1 or 2, related to occurred natural hazards) could give more value to the entire work, otherwise, the result will be just a list of points. I understand the limited text requested for this Special Issue, but maybe 1-2 sentences per section on practical examples could be acceptable by the Editors.

Reply: The lack of practical examples was also raised by the other reviewer and we have thus added some more concrete illustrations throughout the revised manuscript. These include e.g., descriptions of recent activities on scientific coordination with respect to the German flood disaster in 2021, on fostering European exchange in collaboration with platforms from Austria and France, and serious gaming to communicate challenges of crisis management to a variety of stakeholders as well as the general public.

c) one of the future challenges, as mentioned, is communication with the public. Can you enlarge this topic a little? Is it possible to learn some experience from DKKV? The

authors should note that the purpose of this invited paper is to present to our readers the DKKV experience, with useful findings, solutions, criticism, and future challenges; therefore what is the lesson that our reader could learn from DKKV?

Reply: We have added a section on our application of serious gaming to highlight the issue of communication.

**Comments by reviewer 2**

Line 15: How are students involved? Which are the DKKV activities they could participate in?

Reply: Students and recent graduates can apply for a "Young Professional" membership at a significantly reduced annual fee. They can then join all activities of DKKV just like regular members. However, only three YP members can join the annual members' meeting and have a vote there.

We decided to omit these regulatory details in the manuscript as this refers to the statutes of DKKV as an association under German law (i.e. "eingetragener Verein") and does not extend on our activities in the field of disaster risk reduction.

Line 21: How is the public informed? Via a webpage, app or workshops? Some examples would help the reader to get a better understanding of the DKKV.

Reply: We have added examples of our communication activities in the revised manuscript.

These include our website, but also flyers, public lectures, and workshops. Of course, we have been doing that in a digital form during the pandemic.

At the end of the manuscript, we have added a short section on serious gaming; this has proven to be a very successful way of bringing people together and to facilitate the exchange of knowledge.

Line 27: What is the aim of these research projects? (e.g. are they addressing the reduction of climate risk and other hazards?)

Reply: We have already replied to this in the "reply to editor comments" section. The following section was added to the revised manuscript.

*"Within research projects, DKKV generally aims to contribute to communication and dissemination as well as to tasks which require to synthesise research results for different stakeholders and communities. At the same time, these project participations allow DKKV to extend and strengthen its network, develop new methods and approaches (such as serious games), and provide funding for activities that create synergies between project tasks and activities that lie in the field of our institutional goals."*

Line 31 ff : What concrete risks are humanity facing? And why are the conditions more challenging?

Reply: We have slightly changed the sentence (present tense instead of future tense in the first part).

"Humanity is increasingly facing multiple risks under more challenging conditions. The continuation of climate change and the ever more frequent occurrence of extreme, multi-hazard, and cascading events are interacting with increasingly complex and interconnected societies. Natural and technological hazards leading to economic and non-economic losses call for further research to assist policy makers and public administration with evidence-based, precautionary, and actionable information and advice."

Thus, our argument is that the continuation of climate change and the more complex hazard situations interact with more connected and complex vulnerable societies. As both the hazards as well as the objects at risk are becoming more complex, we are convinced that this relates to a more challenging situation than in previous decades.

Line 50: What data should be improved? Why is it not homogeneous? How should the data be homogenized? Please provide more information here.

Reply: Data collection is currently being carried out by a multitude of local, regional, national international institutions from both science and administration. If such data are not harmonized to a certain degree (e.g. data formats, update frequencies, data collected), they hinder the application risk and impact assessments. These are however necessary to assess future impacts and losses and form the basis for risk-informed decision making.

We have changed the sentence in the revision to highlight the need for harmonization of data collection.

Line 51: You are addressing simulations and assessments. Please provide a concrete example to make it more vivid and tangible to the reader.

Reply: We have added some examples to the revised manuscript which focus on impact assessments for droughts (Peichl et al. 2021), digital twins in an urban setting (Losier et a. 2019) as well as a railway system (Kaewunruen et al. 2021).

Line: 69: What does "Build Back Better" mean in practice? Can you provide an example?

Reply: After the occurrence of major extreme events, there is a tendency to aim at building back to the pre-event conditions. We have recently experienced this again after the German flood disaster in 2021. However, to reconstruct sustainably, future risks and their impacts must be known and be accounted for. Scientific guidance is required.

Thus, two major research projects to guide a sustainable reconstruction have been initiated after the flood events by the German Ministry of Research and Education. We briefly describe these in the revised manuscript.

Line 71: Is there an example for an early warning system that you can suggest to the reader?

Reply: There are many best-practices in relation to early warning systems; however, we do not think that there is one that addresses all parts of an early warning chain to a perfect degree, which would also be applicable to all hazards everywhere.

For the purpose of this article, we have added to recent changes in Germany: 1. Legal changes to allow the national government to provide coordination in regional and state-level disaster events, and 2. the upcoming implementation of cell broadcasting for public alerts.

---

## Referee Report (RR1)

**Review of nhess-2021-197-manuscript-version3**
**Authors**: Benni Thiebes et al.
**Title**: "Invited perspectives: Challenges and step changes for natural hazards - perspectives from the German Committee for Disaster Reduction (DKKV)"
**Recommendation**: Acceptable pending minor revisions

**Overview**

The authors have adequately answered my questions and remarks from the previous review cycle and provided additional information and examples in the manuscript. The provided examples resulted in a better understanding on the activities and aims of DKKV. However, I don't think it would hurt to add even more additional descriptions of practical examples/stories if length constraints allow as this would make the content of the manuscript more vivid and tangible to the reader. The added figure makes it easier for the reader to get an overview of the main working fields of DKKV's work (although it is a table rather than a figure). The added table is difficult to read and could perhaps be improved to become more reader friendly.

Overall, the work is well written and meets the objectives of the special issue. I would recommend publication after some minor adjustments.

**Minor remarks:**

Line 51:
I suggest to change the sentence to:
"Such bottom-up approaches  **are an** important mean to facilitate inclusive, integrative, transdisciplinary, and synergetic collaboration."

Table 1:
I find the table difficult to read and a bit confusing. Could you add a little more description to the caption. Are the step changes described in the different lines of the table? Some points start with capital letters, others not. Is this done on purpose?

Line 105:
This link looks a bit messy with the "test" and "Draft" in it. I am not sure if using compressed links (e.g. https://rb.gy/dffmab in this case) meets the requirements of the journal but it could be worth checking.

Line 108:
Start reference Fleming et al. 2020 in  new line

Line 113:
Start reference Losier et al. 2019 in new line. This will also repair the link to Kaewunruen et al. (2021), which is currently not working due to the missing space.

Line 127:
I clicked on the url provided and received a bad request. Copying the full url into a browser did work. Did the line break prohibited the link to open?

---

## Author Response (AR2)

Dear editor, dear reviewers,

Many thanks for your feedback on our revised manuscript.
We have made the final corrections as requested by you.

Best regards
Benni Thiebes and the other authors

Minor remarks:

Line 51: I suggest to change the sentence to: "Such bottom-up approaches represent and are an important mean to facilitate inclusive, integrative, transdisciplinary, and synergetic collaboration."
**Reply:** Changed as suggested.

Table 1: I find the table difficult to read and a bit confusing. Could you add a little more description to the caption. Are the step changes described in the different lines of the table? Some points start with capital letters, others not. Is this done on purpose?
**Reply:** We have changed all lines to lower-case letters and have also used this style for the figure 1. We have changed the caption to: "Table 1: Step changes and research needs along the priorities of the SFDRR.". We have also added bullet points to the table make reading easier. (the original multi-line table was meant to make formatting and reading easier, however, your comments shows that this was rather confusing).

Line 105: This link looks a bit messy with the "test" and "Draft" in it. I am not sure if using compressed links (e.g. https://rb.gy/dffmab in this case) meets the requirements of the journal but it could be worth checking.
**Reply:** We agree that the citation and the link are messy. We have checked the web and found the final version of the report. We have thus changed the citation in the text as well as in the list of references.

Line 108: Start reference Fleming et al. 2020 in new line
**Reply:** Changed as suggested.

Line 113: Start reference Losier et al. 2019 in new line. This will also repair the link to Kaewunruen et al. (2021), which is currently not working due to the missing space.
**Reply:** Changed as suggested.

Line 127: I clicked on the url provided and received a bad request. Copying the full url into a browser did work. Did the line break prohibited the link to open?
**Reply:** We think that the "." at the end caused the link to not work properly. We have changed this.